# Extracorporeal Membrane Oxygenation for Severe Acute Respiratory Distress Syndrome: Propensity Score Matching

**DOI:** 10.3390/membranes11060393

**Published:** 2021-05-26

**Authors:** Li-Chung Chiu, Li-Pang Chuang, Shaw-Woei Leu, Yu-Jr Lin, Chee-Jen Chang, Hsin-Hsien Li, Feng-Chun Tsai, Chih-Hao Chang, Chen-Yiu Hung, Shih-Wei Lin, Han-Chung Hu, Chung-Chi Huang, Huang-Pin Wu, Kuo-Chin Kao

**Affiliations:** 1Department of Thoracic Medicine, Chang Gung Memorial Hospital, Chang Gung University College of Medicine, Taoyuan 33305, Taiwan; pomd54@cgmh.org.tw (L.-C.C.); r5243@adm.cgmh.org.tw (L.-P.C.); swleu@cgmh.org.tw (S.-W.L.); ma7384@adm.cgmh.org.tw (C.-H.C.); simamura@cgmh.org.tw (C.-Y.H.); ec108146@adm.cgmh.org.tw (S.-W.L.); h3226@cgmh.org.tw (H.-C.H.); cch4848@cgmh.org.tw (C.-C.H.); 2Graduate Institute of Clinical Medical Sciences, College of Medicine, Chang Gung University, Taoyuan 33302, Taiwan; 3Department of Thoracic Medicine, New Taipei Municipal TuCheng Hospital, Chang Gung University, Taoyuan 33302, Taiwan; 4Research Services Center for Health Information, Chang Gung University, Taoyuan 33302, Taiwan; doublelin15@gmail.com (Y.-J.L.); cjchang@mail.cgu.edu.tw (C.-J.C.); 5Clinical Informatics and Medical Statistics Research Center, Chang Gung University, Taoyuan 33302, Taiwan; 6Department of Respiratory Therapy, Chang Gung University College of Medicine, Taoyuan 33302, Taiwan; hsinhsien@mail.cgu.edu.tw; 7Institute of Emergency and Critical Care Medicine, School of Medicine, National Yang Ming Chiao Tung University, Taipei 11221, Taiwan; 8Division of Cardiovascular Surgery, Chang Gung Memorial Hospital, Taoyuan 33305, Taiwan; lutony@cgmh.org.tw; 9Department of Respiratory Therapy, Chang Gung Memorial Hospital, Chang Gung University College of Medicine, Taoyuan 33305, Taiwan; 10Division of Pulmonary, Critical Care and Sleep Medicine, Chang Gung Memorial Hospital, Keelung 20401, Taiwan; whanpyng@cgmh.org.tw

**Keywords:** acute respiratory distress syndrome, extracorporeal membrane oxygenation, propensity score matching analysis, mortality

## Abstract

The high mortality rate of patients with severe acute respiratory distress syndrome (ARDS) warrants aggressive clinical intervention. Extracorporeal membrane oxygenation (ECMO) is a salvage therapy for life-threatening hypoxemia. Randomized controlled trials of ECMO for severe ARDS comprise a number of ethical and methodological issues. Therefore, indications and optimal timing for implementation of ECMO, and predictive risk factors for outcomes have not been adequately investigated. We performed propensity score matching to match ECMO-supported and non-ECMO-supported patients at 48 h after ARDS onset for comparisons based on clinical outcomes and hospital mortality. A total of 280 severe ARDS patients were included, and propensity score matching of 87 matched pairs revealed that the 90-d hospital mortality rate was 56.3% in the ECMO group and 74.7% in the non-ECMO group (*p* = 0.028). Subgroup analysis revealed that greater severity of ARDS, higher airway pressure, or a higher Sequential Organ Failure Assessment score tended to benefit from ECMO treatment in terms of survival. Multivariate logistic regression revealed that hospital mortality was significantly lower among patients who received ECMO than among those who did not. Our findings suggested that early initiation of ECMO (within 48 h) may increase the likelihood of survival for patients with severe ARDS.

## 1. Introduction

Acute respiratory distress syndrome (ARDS) is a lethal form of acute respiratory failure with hypoxemia. The mortality rate among patients with the severe form of ARDS exceeds 40% [1,2]. A lung-protective ventilation strategy with lower tidal volumes and lower airway pressures remained the mainstay of management for ARDS, and early application of prolonged prone position was suggested for severe ARDS to improve oxygenation and improve the likelihood of survival [3,4]. In the event that lung-protective ventilation is ineffective, extracorporeal membrane oxygenation (ECMO) can be used as a rescue therapy to treat refractory hypoxemia and alleviate ventilator-induced lung injury [5,6,7]. However, ECMO is a highly invasive technique with potentially devastating complications, and it is generally applied only in highly enabled medical centers [5,7,8]. At present, the actual survival benefits of ECMO in cases of severe ARDS have not been thoroughly investigated.

Two recent randomized controlled trials have assessed the efficacy of ECMO in cases of severe ARDS: Conventional Ventilatory Support versus Extracorporeal Membrane Oxygenation for Severe Adult Respiratory Failure (CESAR) [9] and the ECMO to Rescue Lung Injury in Severe ARDS (EOLIA) [10]. Trials of ECMO as a rescue treatment in patients with severe ARDS pose difficult ethical and methodological issues. Nonetheless, propensity score matching can be used in observational cohort studies to elucidate the effects of ECMO in patients with severe ARDS in real-world scenarios [11].

In the current study, we applied propensity score matching to data obtained from previous observational cohort studies performed in a high-volume ECMO center in Taiwan [12,13]. Our primary objective was to elucidate the influence of ECMO on clinical outcomes and hospital mortality among patients with severe ARDS. Specifically, we sought to determine whether the 28-day, 60-day, or 90-day in-hospital mortality of ECMO patients was lower than that of non-ECMO patients. We also compared the two groups in terms of ventilator-, ICU-, and hospital-free days.

## 2. Materials and Methods

### 2.1. Study Design and Patients

This matched cohort study was based on propensity scores derived from previous studies of patients with severe ARDS. One study, conducted between May 2006 and October 2015 [12], collected background characteristics, variables of arterial blood gas, mechanical ventilator settings, and Sequential Organ Failure Assessment (SOFA) scores obtained prior to ECMO initiation. Another prospective observational cohort study conducted between September 2012 and September 2015 [13] collected background characteristics, variables of arterial blood gas, mechanical ventilator settings, and the SOFA score on the day of ARDS onset (day 1) as well as on days 3 (i.e., 48 h), 7, and 14 after the initial diagnosis. All severe ARDS patients with or without ECMO support were deeply sedated and paralyzed with continuous neuromuscular blockade during the first days of ARDS, and respiratory mechanics data were collected during neuromuscular blockade.

Both of these studies were conducted at Chang Gung Memorial Hospital (CGMH) in Taiwan, a tertiary care referral center featuring a 3700-bed general ward and 278-bed adult intensive care unit (ICU). It also operates as an ECMO center, with more than 100 cases of venoarterial and venovenous ECMO annually. At this institution, the decision to initiate ECMO cannulation is made by the treating intensivist and cardiac surgeon. The criteria for ECMO initiation in severe ARDS patients were persistent hypoxemia (PaO_2_/FiO_2_ ratio <80 mm Hg) for at least 6 h despite aggressive mechanical ventilation support as positive end-expiratory pressure (PEEP) >10 cm H_2_O or peak inspiratory pressure >35 cm H_2_O. The exclusion criteria were (1) age <20 years, (2) malignancies with poor prognosis within 5 years, (3) significant underlying comorbidities or severe multiple organ failure refractory to treatment, and (4) mortality within 24 h after ECMO initiation. The initial mechanical ventilator settings protocol after ECMO support was as follows: tidal volume 4–6 mL/kg predicted body weight; PEEP 10–15 cm H_2_O; peak inspiratory pressure 25–30 cm H_2_O; respiratory rate 10–12 breaths per minute; and FiO_2_ adjusted to maintain arterial oxygen saturation above 90%. The ECMO introduction criteria and mechanical ventilator settings protocol did not change from 2006 to 2015.

In recent studies on severe ARDS patients with refractory hypoxemia, the median ARDS duration prior to ECMO initiation has been 2 days [9,14,15,16]. Thus, we recruited non-ECMO patients who were still diagnosed with severe ARDS at 48 h after ARDS onset. This approach was meant to control for potential bias, which may have arisen when matching patients who received ECMO at a later time point with those who did not receive ECMO at baseline or an earlier time point. The local Institutional Review Board for Human Research approved this study (CGMH IRB No. 201600632B0 and 201203949B0) and waived the need for informed consent.

### 2.2. Definitions

Severe ARDS was defined in accordance with the Berlin criteria [1]. Hospital mortality refers to all-cause death during the hospital stay. Patients who remained alive for 90 days after discharge from the hospital were regarded as survivors. Dynamic driving pressure was calculated as peak inspiratory pressure minus PEEP [12,17], and mechanical power was calculated using the following equation [17,18,19]:

Mechanical power (Joules/minutes) (J/min) = 0.098 × tidal volume × respiratory rate × (peak inspiratory pressure − 1/2 × driving pressure).

### 2.3. Statistical Analysis

Continuous variables were presented as mean ± standard deviation or median (interquartile range), and categorical variables were reported as numbers (percentages). The student’s *t* test or the Mann-Whitney *U* test was used to compare continuous variables between groups. Categorical variables were tested using the chi-square test for equal proportions or Fisher’s exact test.

Propensity score matching was performed to reduce the confounding effects and the likelihood of selection bias using a nearest-neighbor algorithm with 1:1 matching without replacement and a caliper distance of less than 0.2. Matching was performed for age, body mass index, SOFA score, PaO_2_/FiO_2_ ratio, and dynamic driving pressure. Each ECMO-supported patient was matched with a non-ECMO supported patient presenting the smallest absolute difference in propensity scores. Receiver operating characteristic curves and the Youden index were used to determine the cutoff to dichotomize continuous variables. For subgroup analysis, treatment and subgroup interactions were tested using a one-step model with median values to partition subgroups based on quantitative characteristics. Multivariate logistic regression was also used to estimate the probability of receiving ECMO for each observation. Risk factors associated with hospital mortality were analyzed using univariate analysis in the first step, followed by a multivariate logistic regression model with stepwise selection. The results are presented using odds ratio (OR) and 95% confidence intervals (CI). Cumulative mortality curves were generated as a function of time using the Kaplan-Meier approach and compared using the log-rank test. All statistical analysis was performed using SPSS 22.0 and R 3.6.3 statistical software, and a two-sided *p* value of <0.05 was considered statistically significant.

## 3. Results

### 3.1. Patients

The ECMO group was drawn from a pool of 158 patients with severe ARDS who received ECMO (between 2006 and 2015), and the overall all-cause in-hospital mortality rate was 55.1%. The non-ECMO study group was drawn from 122 patients who were reclassified as severe ARDS at 48 h after ARDS onset (between 2012 and 2015), and the overall all-cause in-hospital mortality rate was 79.5%. Nearly all of the severe ARDS patients received pressure-controlled ventilation, regardless of whether they received ECMO support. Procedural implementation of ECMO did not vary significantly between 2006 and 2015, and we observed no significant variations in hospital mortality throughout the study period (2006–2011: mortality rate 55.7%; 2012–2015: mortality rate 54.4%, *p* = 0.873). We included 87 patients within each group, based on propensity score matching for ARDS severity and patient characteristics (Figure 1). 

### 3.2. Comparisons of ECMO Group and non-ECMO Group before and after Matching

Prior to matching, age, body mass index, and SOFA scores were higher in the non-ECMO group than in the ECMO group (all *p* < 0.05) (Table 1). We observed no significant differences between ECMO and non-ECMO patients in terms of comorbidities or any of the baseline ventilator settings other than dynamic driving pressure (*p* = 0.022). Hospital mortality was significantly lower in the ECMO group than in the non-ECMO group (55.1% vs. 79.5%, *p* < 0.001).

After matching, the median ARDS duration prior to ECMO initiation was 40 h in the ECMO group, which is close to the ARDS duration at the time when the patient was reclassified as severe ARDS in the non-ECMO group (i.e., 48 h after ARDS onset). We observed no significant differences between the two groups in terms of demographics, comorbidities, or clinical variables (including ventilator settings). Nonetheless, hospital mortality was significantly lower among patients in the ECMO group than among those in the non-ECMO group (59.8% vs. 77%, *p* = 0.014).

### 3.3. Outcomes

Prior to matching, the overall 90-day survival rate was significantly higher among patients with ECMO support than among those without ECMO support (47.5% vs. 22.1%, *p* < 0.001, log-rank test) (Figure 2a). After matching, the overall 90-day survival rate was still significantly higher among patients with ECMO support than among those without ECMO support (43.7% vs. 25.3%, *p* = 0.028, log-rank test) (Figure 2b). Survival curves after matching revealed a higher prevalence of early death in the ECMO group; however, their 28-day hospital mortality rate was non-significantly lower (*p* = 0.371), and the 60-day hospital mortality rate was significantly lower (*p* = 0.025). The duration of mechanical ventilation, length of ICU stay, and hospital stay were all significantly longer in the ECMO group (all *p* < 0.05). No significant differences were observed between the two groups in terms of the number of ventilator-free days at 28, 60, or 90 days. The number of 90-day ICU-free days was significantly longer in the ECMO group (*p* = 0.046) (Table 2).

### 3.4. Subgroup Analysis

As shown in Figure 3, patients with ARDS of greater severity (lung injury score > 3.5, PaO_2_/FiO_2_ ≤ 75 mmHg), higher airway pressure (PEEP > 12 cm H_2_O, peak inspiratory pressure > 33 cm H_2_O, or dynamic driving pressure > 20.5 cm H_2_O), or organ dysfunction of greater severity (SOFA score > 11) benefited significantly from ECMO treatment (all *p* < 0.05).

### 3.5. Multivariate Adjustment

After adjusting for possible confounders in multivariate logistic regression analysis, the risk of death was significantly lower in the ECMO group than in the non-ECMO group (Adjusted OR 0.40 (95% CI 0.19–0.81); *p* = 0.013). Higher SOFA scores and higher dynamic driving pressure were independently associated with higher hospital mortality, regardless of whether the patient received ECMO support (Adjusted OR 1.25 (95% CI 1.11–1.43); *p* < 0.001, and Adjusted OR 1.18 (95% CI 1.02–1.39); *p* = 0.038, respectively) (Table 3).

## 4. Discussion

Our primary insight in this research was the fact that after propensity score matching, the 90-day hospital mortality of patients who received ECMO was significantly lower than that of patients who did not receive ECMO support.

The severity of ARDS can change considerably during first few days after onset, and cases of severe ARDS confirmed after 24 to 48 h present a higher risk of mortality [2,13]. In the LUNG SAFE study, the mortality rate at the time of ARDS onset (43%) was lower than in cases reclassified as severe ARDS after 24 h (57%) [2]. As compared to patients in the LUNG SAFE study, those in the current study were older, presented more chronic diseases, and received higher airway pressure. This may explain why the mortality rates in the current study were much higher: ARDS onset (58.6%) and reclassified as severe ARDS after 48 h without ECMO support (79.5%). Our findings suggest that patients with severe ARDS form a high-risk subgroup for whom adjunctive interventions should be considered to treat profound hypoxemia.

Two recent randomized controlled trials of ECMO for patients with severe ARDS presented conflicting results [9,10]. The CESAR trial reported that transfer to an ECMO center was preferable to continued conventional ventilation in terms of 6-month survival without disability. However, 24% of the patients randomly assigned to the ECMO arm did not actually undergo ECMO, and 30% of the patients in the control arm did not actually receive lung-protective ventilation [10]. Another EOLIA trial reported that 60-day mortality in the ECMO group was not significantly lower than in the non-ECMO control group. However, the trial was stopped early, and 28% of the patients in the control group crossed over to the ECMO group.

Trials of ECMO involving patients with severe ARDS pose difficult methodological and ethical issues. Strict exclusion criteria based on patient characteristics are not necessarily feasible in real-world scenarios, such that the recruitment of a sufficient number of patients would take a long time in the ECMO trials. Note that the CESAR and EOLIA trials required more than five years to assemble a meaningful cohort [9,10]. Another issue is guaranteeing the safe transport of patients and balancing the risks versus the benefits of referral to specialized medical centers. There is the issue of variability among centers in terms of ECMO techniques and patient management. There is also the issue of crossover from the control group to ECMO treatment [5,20]. Taken together, these issues make it exceedingly difficult to interpret the results of ECMO trials or form informed conclusions pertaining to the actual benefits of ECMO in real-world scenarios.

One previous study using propensity score matching for influenza-induced ARDS reported similar mortality rates among patients treated using ECMO and those receiving conventional interventions. However, those patients were suffering from ARDS of homogenous etiology (i.e., pandemic influenza A; H1N1), which is not necessarily generalizable to ARDS from other causes [14]. Thus, we applied propensity score matching to the findings obtained in previous observational cohort studies covering a range of etiologies [12,13]. Our results revealed that after matching, 60-day and 90-day hospital mortality was significantly lower in the ECMO group and 90-day ICU-free days was significantly longer. These results are consistent with those of two meta-analysis studies [20,21]. One recent observational cohort study also reported that ECMO reduced mortality in patients with severe respiratory failure due to coronavirus disease 2019 (COVID-19) [16]. These above studies demonstrated that ECMO treatment may have a survival benefit in some patients with severe ARDS.

However, severe ARDS patients supported with ECMO in our study had higher mortality than previous reports [9,10,19,22,23,24,25]; this may be due to the fact that our hospital is a tertiary care referral center, and we did not exclude patients with malignancies or severe comorbidities. Furthermore, a previous study revealed that prone positioning before ECMO was independently associated with lower mortality for severe ARDS patients [23]; however, prone positioning was underutilized in the current study (<5%). Recent studies showed that ventilator settings during ECMO were associated with mortality for severe ARDS patients, and our patients received higher airway pressure, and higher tidal volume during the first days of ECMO than the values in these studies [12,14,24,25], which probably also elevated the mortality rates.

The most favorable timing for ECMO initiation in patients with severe ARDS has yet to be determined [5,26]. After matching patients at 48 h after ARDS onset, our multivariate analysis revealed that ECMO treatment was independently associated with a lower likelihood of death (Adjusted OR = 0.4, *p* = 0.013), based on the fact that the hospital mortality rate in the ECMO group was significantly lower than in the non-ECMO group. As in previous studies, our findings indicate that early ECMO intervention (i.e., not only as a rescue therapy) [5,22], may be beneficial to the survival of severe ARDS patients with profound hypoxemia [5,19,21,22,23].

Researchers have yet to identify the indications for ECMO or formulate predictive mortality risk models conclusively [5,6,7,26]. Severe hypoxemia is a prevalent indicator for the initiation of ECMO in ARDS cases. Subgroup analysis in this study revealed that patients with PaO_2_/FiO_2_ ≤ 75 mmHg or lung injury scores >3.5 would benefit from ECMO. Note however that the severity of ARDS was not independently associated with hospital mortality, as in previous reports [5,19,22,23].

We determined that implementing mechanical ventilation in conjunction with lung-protective strategies can improve survival outcomes in ARDS patients. We also observed a correlation between higher airway pressures and poor outcomes. One previous post hoc observational study concluded that driving pressure was the ventilation variable most strongly associated with survival among ARDS patients [27]. The LUNG SAFE study reported that higher peak inspiratory pressure, plateau pressure, and driving pressure were associated with higher hospital mortality among ARDS patients [28]. Higher airway pressure (i.e., peak inspiratory pressure or plateau pressure) prior to ECMO institution was also shown to be independently predictive of mortality in patients with severe ARDS requiring ECMO [22,23]. This study demonstrated that among patients with higher airway pressure (i.e., PEEP > 12 cm H_2_O, peak inspiratory pressure >33 cm H_2_O, or dynamic driving pressure >20.5 cm H_2_O), the odds ratio of death was lower among those treated with ECMO than among those without ECMO support. Multivariate analysis also revealed that dynamic driving pressure was independently associated with hospital mortality. Taken together, these results indicate that ECMO facilitates ultra-protective ventilation by allowing clinicians to reduce mechanical forces acting on the lungs, thereby mitigating the risk of ventilator-induced lung injury [6,7,18,21,29].

The most common cause of death among ARDS patients is multiorgan failure [30]. The LUNG SAFE study reported a correlation between the degree of systemic organ failure and ARDS outcomes [2]. It has also been reported that organ failure status prior to ECMO initiation is associated with the likelihood of mortality among patients with severe ARDS requiring ECMO [19,21,22,23]. In a previous study, we reported that in patients with severe ARDS, SOFA scores decreased significantly during the first three days of ECMO (after cannulation) with a corresponding impact on hospital mortality [18]. This study reported that the odds ratio of death was lower in patients with SOFA scores >11 when treated with ECMO. Multivariate analysis also revealed that SOFA scores were independently associated with hospital mortality. This may be due to the fact that ECMO allows clinicians to reduce the mechanical power of ventilation to alleviate ventilator-induced lung injury by reducing proinflammatory biotrauma response, thereby preventing further multi-organ failure and improving survival outcomes [6,18,21,29].

This study was hindered by a number of limitations. First, this study was conducted in a single tertiary medical center, such that our results are not necessarily generalizable to other facilities. Note also that our specialized hospital performs a large number of ECMO operations, and familiarity with the procedure may contribute to lower mortality [8]. Nonetheless, this study did not exclude older patients or those with malignancy or chronic comorbidities, which probably elevated the mortality rates. Taken together, it is difficult to generalize our findings to those of other ICUs or hospitals. Second, early application of prolonged prone positioning for severe ARDS patients had a survival benefit noted since 2013 [4]; however, it is not necessarily followed in real-world clinical practice. For example, only 16.3% of patients with severe ARDS received prone positioning in the LUNG SAFE study [31], and fewer than 5% use in the current study, which could influence the clinical outcome and results. Finally, despite our use of propensity score matching to minimize differences in clinical parameters, any number of residual or confounding variables that were not measured or matched could impose unrecognized imbalances in baseline features and thereby influence the results.

## 5. Conclusions

Our findings provide further evidence that among patients with severe ARDS, the 90-day hospital mortality rate was significantly lower among those who received ECMO than among those who did not receive ECMO. Note that this was despite similarities in severity and clinical characteristics within 48 h after ARDS onset, as determined using propensity score matching.

## Figures and Tables

**Figure 1 membranes-11-00393-f001:**
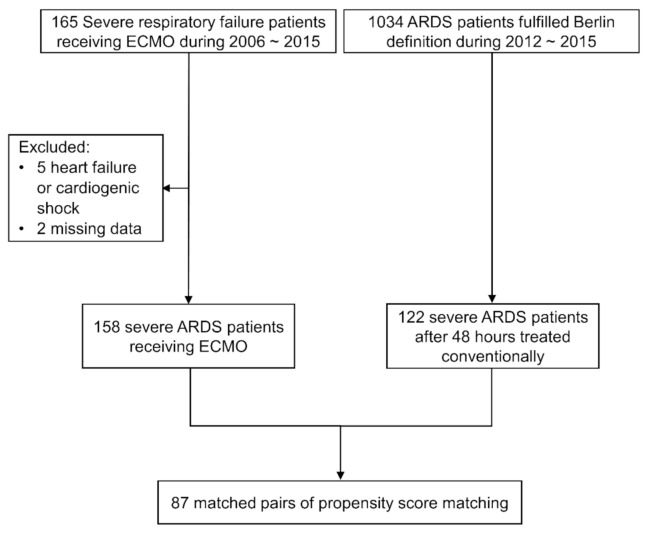
Flow diagram showing the enrollment of patients with severe ARDS who were or were not administered ECMO support via propensity score matching. (ARDS, acute respiratory distress syndrome; ECMO, extracorporeal membrane oxygenation).

**Figure 2 membranes-11-00393-f002:**
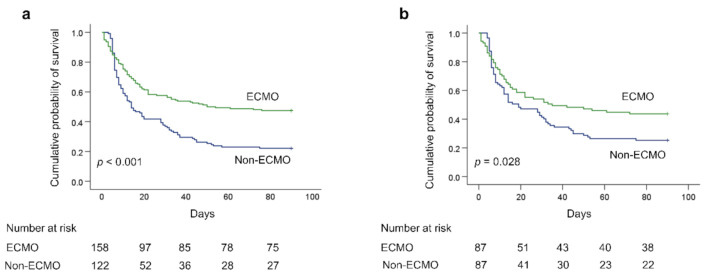
Kaplan-Meier 90-d survival curves of patients who did or did not receive ECMO support for severe acute respiratory distress syndrome: (**a**) before propensity score matching; (**b**) after propensity score matching. (ECMO, extracorporeal membrane oxygenation).

**Figure 3 membranes-11-00393-f003:**
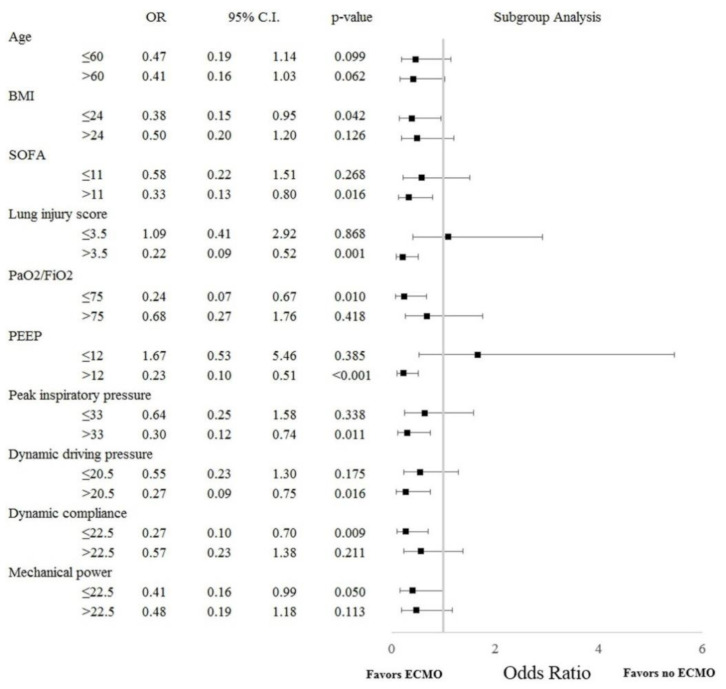
Subgroup analysis of hospital mortality as a function of baseline clinical variables. (BMI, body mass index; CI, confidence interval; ECMO, extracorporeal membrane oxygenation; FiO_2_, fraction of inspired oxygen; OR, odds ratio; PaO_2_, partial pressure of oxygen in arterial blood; PEEP, positive end-expiratory pressure; SOFA, sequential organ failure assessment).

**Table 1 membranes-11-00393-t001:** Background characteristics of severe ARDS patients with or without ECMO before and after matching.

Variables	Before Matching			After Matching		
	ECMO	Non-ECMO		ECMO	Non-ECMO	
	(*n* = 158)	(*n* = 122)	*p*	(*n* = 87)	(*n* = 87)	*p*
Age (years)	50.3 ± 16.3	65.1 ± 14.1	<0.001	58.3 ± 13.2	61.9 ± 14.3	0.082
Gender (male)	108 (68.4%)	79 (64.8%)	0.526	58 (66.7%)	60 (69%)	0.746
Body mass index (kg/m^2^)	25.8 ± 5.2	23.7 ± 4.4	0.001	24.8 ± 4.0	24.2 ± 4.6	0.323
ARDS etiologies						
Pulmonary cause	121 (76.6%)	95 (77.9%)	0.799	66 (75.9%)	69 (79.3%)	0.585
Extrapulmonary cause	37 (23.4%)	27 (22.1%)	0.799	21 (24.1%)	18 (20.7%)	0.585
Comorbidities						
Diabetes mellitus	40 (25.3%)	33 (27%)	0.743	29 (33.3%)	19 (21.8%)	0.09
Cerebrovascular accident	10 (6.3%)	12 (9.8%)	0.280	9 (10.3%)	4 (4.6%)	0.248
Congestive heart failure	13 (8.2%)	5 (4.1%)	0.162	9 (10.3%)	3 (3.4%)	0.132
Coronary artery disease	6 (3.8%)	6 (4.9%)	0.646	6 (6.9%)	4 (4.6%)	0.747
Chronic lung disease	16 (10.1%)	16 (13.1%)	0.436	7 (8%)	14 (16.1%)	0.103
Liver cirrhosis or chronic hepatitis	22 (13.9%)	21 (17.2%)	0.449	16 (18.4%)	17 (19.5%)	0.847
Chronic kidney disease	18 (11.4%)	14 (11.5%)	0.983	14 (16.1%)	9 (10.3%)	0.263
Malignancies	24 (15.2%)	20 (16.4%)	0.784	16 (18.4%)	15 (17.2%)	0.843
SOFA score	10.9 ± 3.2	12.2 ± 3.5	0.001	11.4 ± 3.2	11.8 ± 3.5	0.417
Lung injury score	3.4 ± 0.4	3.4 ± 0.4	0.225	3.4 ± 0.4	3.4 ± 0.4	0.482
ARDS duration before ECMO (h)	28 (7–129)			40 (8–153)		
PaO_2_/FiO_2_ (mm Hg)	81.1 ± 50.4	78.7 ± 17.8	0.576	80.4 ± 45.7	79.2 ± 18.8	0.809
PaCO_2_ (mm Hg)	52.2 ± 18.8	52.8 ± 19.0	0.803	50.6 ± 17.9	53.9 ± 20.4	0.258
Ventilator settings						
Tidal volume (ml/kg PBW)	7.7 ± 2.4	8.2 ± 2.4	0.098	8.2 ± 2.6	8.1 ± 2.4	0.795
PEEP (cm H_2_O)	12.0 ± 2.8	12.5 ± 2.6	0.123	12 ± 2.6	12. 3 ± 2.6	0.428
Peak inspiratory pressure (cm H_2_O)	33.9 ± 6.5	32.8 ± 5.9	0.130	33.1 ± 5.8	33.3 ± 5.4	0.861
Dynamic driving pressure (cm H_2_O)	21.9 ± 6.2	20.3 ± 5.5	0.022	21.0 ± 6.1	20.9 ± 5.0	0.935
Dynamic compliance (ml/cm H_2_O)	22.5 ± 11.2	25.6 ± 16.6	0.072	24.1 ± 12.7	23.8 ± 9.8	0.828
Total respiratory rate (breaths/min)	24.2 ± 7.1	25.2 ± 4.7	0.142	24.2 ± 6.6	25.3 ± 4.8	0.207
Minute ventilation (L/min)	10.6 ± 3.8	10.7 ± 3.2	0.858	10.8 ± 4.0	10.8 ± 3.0	1.000
Mechanical power (J/min)	23.7 ± 9.6	23.6 ± 7.7	0.910	23.8 ± 10.0	23.9 ± 7.0	0.946
MP/PBW (×10^−3^ J/min/kg)	415 ± 172	437 ± 170	0.297	430 ± 188	433 ± 152	0.889
MP/Compliance (J/min/mL/cm H_2_O)	1.27 ± 0.76	1.12 ± 0.54	0.065	1.18 ± 0.66	1.15 ± 0.51	0.751
Duration of mechanical ventilator (days)	20 (11.5–38)	14 (8–28.3)	0.001	20 (12–35.9)	16 (8–30)	0.019
Length of ICU stay (days)	23 (13–43)	16 (8–30)	0.003	23 (15–43)	17 (10–31)	0.031
Length of hospital stay (days)	38.5 (20.8–64)	29.5 (15–51.3)	0.017	37 (20–68)	30 (14–52)	0.035
Hospital mortality, *n* (%)	87 (55.1%)	97 (79.5%)	<0.001	52 (59.8%)	67 (77%)	0.014

Data is presented as mean ± standard deviation, count or median (interquartile range). Abbreviations: ARDS, acute respiratory distress syndrome; ECMO, extracorporeal membrane oxygenation; FiO_2_, fraction of inspired oxygen; ICU, intensive care unit; MP, mechanical power; PaCO_2_, partial pressure of carbon dioxide in arterial blood; PaO_2_, partial pressure of oxygen in arterial blood; PBW, predicted body weight; PEEP, positive end-expiratory pressure; SOFA, Sequential Organ Failure Assessment.

**Table 2 membranes-11-00393-t002:** Clinical outcomes of severe ARDS patients with or without ECMO after matching.

Outcomes	ECMO	Non-ECMO	*p*
	(*n* = 87)	(*n* = 87)	
Mortality			
28 day hospital mortality, *n* (%)	40 (46%)	46 (52.9%)	0.371
60 day hospital mortality, *n* (%)	47 (54%)	64 (73.6%)	0.025
90 day hospital mortality, *n* (%)	49 (56.3%)	65 (74.7%)	0.028
Other outcomes			
Ventilator-free days on day 28	2.8 ± 6.1	2.8 ± 6.0	0.974
Ventilator-free days on day 60	12.8 ± 19.1	9.4 ± 17.5	0.218
Ventilator-free days on day 90	23.9 ± 32.6	16.4 ± 29.8	0.114
ICU-free days on day 90	21.8 ± 29.4	13.3 ± 26.4	0.046
Hospital-free days on day 90	12.9 ± 21.7	9.3 ± 20.1	0.266

Data are presented as mean ± standard deviation, count or median (interquartile range). Abbreviations: ARDS, acute respiratory distress syndrome; ECMO, extracorporeal membrane oxygenation; ICU, intensive care unit.

**Table 3 membranes-11-00393-t003:** Multivariate logistic regression analysis with hospital mortality as outcome.

Factors	Odds Ratio (95% CI)	*p*
ECMO	0.40 (0.19–0.81)	0.013
SOFA score	1.25 (1.11–1.43)	<0.001
Dynamic driving pressure	1.18 (1.02–1.39)	0.038

Abbreviation: CI, confidence interval; ECMO, extracorporeal membrane oxygenation; SOFA, Sequential Organ Failure Assessment. The multivariate analysis model included ECMO, age, body mass index, SOFA, lung injury score, PaO_2_/FiO_2_, positive end-expiratory pressure, peak inspiratory pressure, dynamic driving pressure, dynamic compliance, and mechanical power.

## Data Availability

All data will be available from the corresponding author on reasonable request.

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
