# Peer review of "Extracorporeal Membrane Oxygenation for Severe Acute Respiratory Distress Syndrome: Propensity Score Matching"

_membranes, 2021, doi:10.3390/membranes11060393_

Round 1

Reviewer 1 Report

I congratulate with the authors for the study design and findings.

I have few comments:

1) please specify criteria for running ECMO (not all readers know them). 

2) Age AND BMI between population are statistically different (and clinically relevant the formed). This could be a selection bias.

3) ECMO mortality sounds higher than previously reported data. A deeper discussion is required

Author Response

I congratulate with the authors for the study design and findings.

I have few comments:

Point 1: please specify criteria for running ECMO (not all readers know them).

Response 1: We thank the reviewer’s suggestions to specify criteria for running ECMO.

The decision to initiate ECMO was made by the treating intensivist and cardiac surgeon when persistent hypoxemia (PaO2/FiO2 ratio < 80 mm Hg) at least 6 hours despite aggressive mechanical ventilation support as positive end-expiratory pressure (PEEP) > 10 cm H2O or peak inspiratory pressure > 35 cm H2O. The exclusion criteria were (1) age < 20 years (2) malignancies with poor prognosis within 5 years (3) significant underlying comorbidities or severe multiple organ failure refractory to treatment (4) mortality within 24 hours after ECMO initiation.

We added the above statement in the second paragraph of Materials and Methods section in the revised manuscript.

Point 2: Age AND BMI between population are statistically different (and clinically relevant the formed). This could be a selection bias.

Response 2: We thank the reviewer to point out this problem that age and BMI between populations are statistically different, and this could be a selection bias.  

In the current study, we applied propensity score matching to elucidate the influence of ECMO on clinical outcomes and hospital mortality among patients with severe ARDS. Before matching, age, body mass index, SOFA scores, and dynamic driving pressure are statistically different between the ECMO group and the non-ECMO group (all p < 0.05) (Table 1). Matching was performed for age, body mass index, SOFA score, PaO2/FiO2 ratio, and dynamic driving pressure (the second paragraph of Statistical analysis section).

After matching, age and BMI between population are not statistically different (58.3 ± 13.2 years versus 61.9 ± 14.3 years, p = 0.082; 24.8 ± 4.0 versus 24.2 ± 4.6, p = 0.323, respectively).

Point 3: ECMO mortality sounds higher than previously reported data. A deeper discussion is required

Response 3: We thank the reviewer to point out that ECMO mortality was higher in our study than previously reported data.

The 90 day hospital mortality rate of patients supported with ECMO in our study was 52.5%. A recent meta-analysis of individual patient data of two randomized controlled trials (i.e., the CESAR and EOLIA trials) [1, 2] showed that 90-day mortality was 36 % for severe ARDS patients supported with ECMO [3]. The largest international, multicenter, prospective cohort study of patients with ARDS receiving ECMO (the LIFEGARDS study) demonstrated that 6-month mortality rate was 39% [4].

The CESAR trial excluded severe ARDS patients with higher peak inspiratory pressure > 30 cm H2O or high FiO2 > 0.8 ventilation for more than 7 days, and the EOLIA trial excluded patients receiving mechanical ventilation longer than 7 days, long-term chronic respiratory insufficiency, cancer with a life expectancy of less than 5 years, moribund condition or greater severity of illness (SAPS-II score more than 90) [1, 2]. Recent prospective or retrospective cohort studies of ECMO also excluded ARDS patients with chronic respiratory failure or other severe comorbidities [4, 5].

Our study enrolled all severe ARDS patients receiving ECMO, and we did not exclude patients receive higher airway pressure or higher FiO2 ventilation for more than 7 days. Furthermore, our hospital is the tertiary care referral center in Taiwan and we also did not exclude patients with malignancy and severe comorbidities, such as chronic heart failure, advanced liver disease, chronic respiratory failure or chronic kidney diseases in the present study, which may contribute to higher mortality in our study.

Early application of prolonged prone position for severe ARDS patients as rescue therapy had survival benefit since 2013 [6], and a previous study revealed that prone positioning before ECMO was independently associated with lower mortality [7]. However, prone positioning were underutilized in the current study (< 5%), which may contribute to higher mortality, and we had already mentioned the limitation of underutilization of prone positioning in the last paragraph of Discussion section in the manuscript.

 ECMO facilitates an ultra-protective ventilation of lowering delivered tidal volume and airway pressure for resting the lungs, and this ultra-protective lung strategy ideally may improve outcomes or mortality by further minimizing ventilator-induced lung injury. Previous studies showed that ventilator settings during ECMO were associated with mortality for patients with severe ARDS [5, 8].

As compared to the recent studies, our patients received higher airway pressure, and higher tidal volume during the first days of ECMO support, which may also contribute to higher mortality [4, 5, 8, 9]. For example, driving pressure and tidal volume during first days of ECMO in our study was 19.8 cmH2O and 6.0 ml/kg predicted body weight, which were higher than the values of 13.7 cmH2O and 4.0 ml/kg predicted body weight in the previous study [8].

These above factors may cause higher mortality in our study. We added the above statement in the sixth paragraph of Discussion section in the revised manuscript.

References:

  1. Peek, G.J.; Mugford, M.; Tiruvoipati, R.; Wilson, A.; Allen, E.; Thalanany, M.M.; Hibbert, C.L.; Truesdale, A.; Clemens, F.; Cooper, N.; et al. CESAR trial collaboration. Efficacy and economic assessment of conventional ventilatory support versus extracorporeal membrane oxygenation for severe adult respiratory failure (CESAR): a multicentre randomised controlled trial. Lancet. 2009, 374, 1351-63.
  2. Combes, A.; Hajage, D.; Capellier, G.; Demoule, A.; Lavoué, S.; Guervilly, C.; Da, Silva. D.; Zafrani, L.; Tirot, P.; Veber, B.; EOLIA Trial Group, REVA, and ECMONet. Extracorporeal Membrane Oxygenation for Severe Acute Respiratory Distress Syndrome. N Engl J Med. 2018, 378, 1965-1975.
  3. Combes, A.; Peek, G.J.; Hajage, D.; Hardy, P.; Abrams, D.; Schmidt, M.; Dechartres, A.; Elbourne, D. ECMO for severe ARDS: systematic review and individual patient data meta-analysis. Intensive Care Med. 2020, 46, 2048-2057.
  4. Schmidt, M.; Pham, T.; Arcadipane, A.; Agerstrand, C.; Ohshimo, S.; Pellegrino, V.; Vuylsteke, A.; Guervilly, C.; McGuinness, S.; Pierard, S.; et al. Mechanical Ventilation Management during Extracorporeal Membrane Oxygenation for Acute Respiratory Distress Syndrome. An International Multicenter Prospective Cohort. Am J Respir Crit Care Med. 2019, 200, 1002-1012.
  5. Schmidt, M.; Stewart, C.; Bailey, M.; Nieszkowska, A.; Kelly, J.; Murphy, L.; Pilcher, D.; Cooper, D.J.; Scheinkestel, C.; Pellegrino, V.; et al. Mechanical ventilation management during extracorporeal membrane oxygenation for acute respiratory distress syndrome: a retrospective international multicenter study. Crit Care Med. 2015, 43, 654-64.
  6. Guérin, C.; Reignier, J.; Richard, J.C.; Beuret, P.; Gacouin, A.; Boulain, T.; Mercier, E.; Badet, M.; Mercat, A.; Baudin, O.; et al. PROSEVA Study Group. Prone positioning in severe acute respiratory distress syndrome. N Engl J Med. 2013 , 368, 2159-68.
  7. Schmidt, M.; Zogheib, E.; Rozé, H.; Repesse, X.; Lebreton, G.; Luyt, C.E.; Trouillet, J.L.; Bréchot, N.; Nieszkowska, A.; Dupont, H.; et al. The PRESERVE mortality risk score and analysis of long-term outcomes after extracorporeal membrane oxygenation for severe acute respiratory distress syndrome. Intensive Care Med. 2013, 39, 1704-13.
  8. Neto, A.S.; Schmidt, M.; Azevedo, L.C.; Bein, T.; Brochard, L.; Beutel, G.; Combes, A.; Costa, E.L.; Hodgson, C.; Lindskov, C.; et al. ReVA Research Network and the PROVE Network Investigators. Associations between ventilator settings during extracorporeal membrane oxygenation for refractory hypoxemia and outcome in patients with acute respiratory distress syndrome: a pooled individual patient data analysis: Mechanical ventilation during ECMO. Intensive Care Med. 2016, 42, 1672-1684.
  9. Chiu, L.C.; Hu, H.C.; Hung, C.Y.; Chang, C.H.; Tsai, F.C.; Yang, C.T.; Huang, C.C.; Wu, H.P.; Kao, K.C. Dynamic driving pressure associated mortality in acute respiratory distress syndrome with extracorporeal membrane oxygenation. Ann Intensive Care. 2017, 7, 12. 

We thank the reviewer for valuable comments. Addressing them fully has significantly strengthened the manuscript.

Reviewer 2 Report

In this study Chiu et al. performed a propensity score matching analysis in two cohorts of severe ARDS patients, and found that 90 days mortality was lower in patients who underwent ECMO treatment. The study was performed in a high volume tertiary center with high ECMO expertise.

The study is well conceived, the results are clearly presented and the discussion is reasonable.

However, I think that it needs some improvements and some points should be clarified:

  • The two cohorts are from slightly different periods (2006-2015 and 2012-2015). The authors declare no changes in ECMO implementation and in hospital mortality for severe ARDS in these 2 periods. Did ECMO introduction criteria and/or mechanical ventilation protocols changed during this 9-years period?

  • The matched cohorts were very similar in gas exchanges, PEEP, driving pressures, mechanical power, minute ventilation. Which patient selection criteria for ECMO were used other than disease severity? Did the matched cohorts differ for these criteria? One may wonder why patients with similar disease have received different treatments.

  • The authors report no data about prone positioning, which is considered a standard of care in moderate-severe and severe ARDS. Do the matched cohorts differ in prone positioning application and response?

  • Methods: line 106. “ Peak inspiratory pressure is equivalent to plateau pressure in pressure-controlled ventilation”. I disagree with this statement. Indeed, it is not very rare to find a difference up to 2-3 cmH2O between Peak and plateau pressure also in pressure controlled mode in patient with moderate-high airway resistance. Moreover, in patients with active inspiratory efforts, Plateau pressure during inspiratory pause can be also higher than peak pressure. Do patients were on neuromuscular blockade during the first days of ARDS? I understand that, due to the retrospective nature of the study, it could be difficult to obtain some data, but daily static respiratory mechanics monitoring is a standard of care in severe ARDS. If it is impossible to have reliable end inspiratory plateau pressures measurements, I suggest the authors to include a statement in the methods section, and to specify if respiratory mechanics data were collected during neuromuscular blockade.

Author Response

In this study Chiu et al. performed a propensity score matching analysis in two cohorts of severe ARDS patients, and found that 90 days mortality was lower in patients who underwent ECMO treatment. The study was performed in a high volume tertiary center with high ECMO expertise.

The study is well conceived, the results are clearly presented and the discussion is reasonable.

However, I think that it needs some improvements and some points should be clarified:

Point 1: The two cohorts are from slightly different periods (2006-2015 and 2012-2015). The authors declare no changes in ECMO implementation and in hospital mortality for severe ARDS in these 2 periods. Did ECMO introduction criteria and/or mechanical ventilation protocols changed during this 9-years period?

Response 1: We thank the reviewer’s comment. The ECMO introduction criteria and mechanical ventilation protocols in our hospital did not change during the 9-years period.  

The criteria for ECMO initiation in severe ARDS patients in our hospital were persistent hypoxemia (PaO2/FiO2 ratio < 80 mm Hg) at least 6 hours despite aggressive mechanical ventilation support as positive end-expiratory pressure (PEEP) > 10 cm H2O or peak inspiratory pressure > 35 cm H2O.

Initial mechanical ventilator settings protocol after ECMO support were as follows: tidal volume 4-6 ml/kg predicted body weight; PEEP 10-15 cm H2O; peak inspiratory pressure 25-30 cm H2O; respiratory rate 10-12 breaths per minute; and FiO2 adjusted to maintain arterial oxygen saturation above 90%. Furthermore, the criteria for weaning from ECMO in our experience were resolving lungs infiltration, lung compliance > 20 ml/cm H2O, PaO> 60 mm Hg and PaCO2  < 45 mm Hg under FiO2 ≦ 0.4, PEEP ≦ 6-8 cm H2O, and peak inspiratory pressure ≦ 30 cm H2O [1].

 We added the ECMO introduction criteria and mechanical ventilation protocols from 2006 to 2015 in the second paragraph of Materials and Methods section in the revised manuscript.

Reference:

  1. Chiu, L.C.; Hu, H.C.; Hung, C.Y.; Chang, C.H.; Tsai, F.C.; Yang, C.T.; Huang, C.C.; Wu, H.P.; Kao, K.C. Dynamic driving pressure associated mortality in acute respiratory distress syndrome with extracorporeal membrane oxygenation. Ann Intensive Care. 2017, 7, 12.

Point 2: The matched cohorts were very similar in gas exchanges, PEEP, driving pressures, mechanical power, minute ventilation. Which patient selection criteria for ECMO were used other than disease severity? Did the matched cohorts differ for these criteria? One may wonder why patients with similar disease have received different treatments.

Response 2: We thank the reviewer to point out these problems. The patient selection criteria for ECMO in severe ARDS in our hospital were persistent hypoxemia (PaO2/FiO2 ratio < 80 mm Hg) at least 6 hours despite aggressive mechanical ventilation support (PEEP > 10 cm H2O or peak inspiratory pressure > 35 cm H2O). The exclusion criteria were (1) age < 20 years (2) malignancies with poor prognosis within 5 years (3) significant underlying comorbidities or severe multiple organ failure refractory to treatment (4) mortality within 24 hours after ECMO initiation.

A total of 158 severe ARDS patients received ECMO between 2006 and 2015 were enrolled, and 87 ECMO-supported patients of them were matched with non-ECMO supported severe ARDS patients based on propensity score matching. Therefore, the matched cohorts did not differ for ECMO selection criteria. After matching, the matched cohorts were very similar in gas exchanges, PEEP, driving pressures, mechanical power, minute ventilation, background characteristics, and other clinical variables.

The precise indications for ECMO in patients with severe ARDS was not clearly defined [1]. Although general patient selection criteria for ECMO were mentioned above, the decision to initiate ECMO was mainly made by the treating intensivist and cardiac surgeon in our hospital.

Reference:

  1. Combes, A.; Schmidt, M.; Hodgson, C.L.; Fan, E.; Ferguson, N.D.; Fraser, J.F.; Jaber, S.; Pesenti, A.; Ranieri, M.; Rowan, K.; et al. Extracorporeal life support for adults with acute respiratory distress syndrome. Intensive Care Med. 2020, 46, 2464-2476.

Point 3: The authors report no data about prone positioning, which is considered a standard of care in moderate-severe and severe ARDS. Do the matched cohorts differ in prone positioning application and response?

Response 3: We thank the reviewer’s suggestion and we agreed that prone positioning is a standard of care in patients with moderate-severe and severe ARDS

 Early application of prolonged prone position for severe ARDS patients as rescue therapy had survival benefit since 2013 [1]. However, underutilization of prone positioning was noted for patients with moderate-severe and severe ARDS in our hospital, even evidenced-based survival benefit of prone positioning was proved after 2013.

Therefore, only a small number of severe ARDS patients underwent prone position in our hospital in real-world clinical practice. Before matching, only two patients in the ECMO group (n = 158) [2], and three patients in the non-ECMO group (n = 122) received prone positioning [3]. Because prone positioning were underutilized in the current study, it is difficult to compare the application and response of prone positioning in the matched cohorts.

 We apologized for not performing comparisons of prone positioning in the matched cohorts due to small sample sizes.

 We had already mentioned the limitation of underutilization of prone positioning in the last paragraph of Discussion section in the manuscript.

Reference:

  1. Guérin, C.; Reignier, J.; Richard, J.C.; Beuret, P.; Gacouin, A.; Boulain, T.; Mercier, E.; Badet, M.; Mercat, A.; Baudin, O.; et al. PROSEVA Study Group. Prone positioning in severe acute respiratory distress syndrome. N Engl J Med. 2013 , 368, 2159-68.
  2. Chiu, L.C.; Hu, H.C.; Hung, C.Y.; Chang, C.H.; Tsai, F.C.; Yang, C.T.; Huang, C.C.; Wu, H.P.; Kao, K.C. Dynamic driving pressure associated mortality in acute respiratory distress syndrome with extracorporeal membrane oxygenation. Ann Intensive Care. 2017, 7, 12.
  3. Chiu, L.C.; Lin, S.W.; Liu, P.H.; Chuang, L.P.; Chang, C.H.; Hung, C.Y.; Li, S.H.; Lee, C.S.; Wu, H.P.; Huang, C.C.; et al. Reclassifying severity after 48 hours could better predict mortality in acute respiratory distress syndrome. Ther Adv Respir Dis. 2020, 14, 1-12.

Point 4: Methods: line 106. “ Peak inspiratory pressure is equivalent to plateau pressure in pressure-controlled ventilation”. I disagree with this statement. Indeed, it is not very rare to find a difference up to 2-3 cmH2O between Peak and plateau pressure also in pressure controlled mode in patient with moderate-high airway resistance. Moreover, in patients with active inspiratory efforts, Plateau pressure during inspiratory pause can be also higher than peak pressure. Do patients were on neuromuscular blockade during the first days of ARDS? I understand that, due to the retrospective nature of the study, it could be difficult to obtain some data, but daily static respiratory mechanics monitoring is a standard of care in severe ARDS. If it is impossible to have reliable end inspiratory plateau pressures measurements, I suggest the authors to include a statement in the methods section, and to specify if respiratory mechanics data were collected during neuromuscular blockade.

Response 4: This is an excellent point of view. We appreciated the reviewer’s comments and agreed that peak inspiratory pressure is not equivalent to plateau pressure in pressure-controlled ventilation. We apologized for this wrong statement and deleted it in the revised manuscript.

All severe ARDS patients with or without ECMO support were deeply sedated and paralyzed with continuous neuromuscular blockade during the first days of ARDS. We added this statement in the first paragraph of Materials and Methods section in the revised manuscript.

We agreed that daily static respiratory mechanics monitoring is a standard of care in severe ARDS, however, end inspiratory plateau pressures was not routinely measured in real-world clinical practices in our hospital. We added a statement “, and respiratory mechanics data were collected during neuromuscular blockade” in the first paragraph of Materials and Methods section in the revised manuscript.

We thank the reviewer for valuable comments. Addressing them fully has significantly strengthened the manuscript.

Round 2

Reviewer 1 Report

no further comments

Reviewer 2 Report

I thank the authors for the accurate response. The manuscript has been improved. I have no other comments.